# *Mycobacterium tuberculosis* YrbE3A Promotes Host Innate Immune Response by Targeting NF-κB/JNK Signaling

**DOI:** 10.3390/microorganisms8040584

**Published:** 2020-04-17

**Authors:** Jieru Wang, Xiaojie Zhu, Yongchong Peng, Tingting Zhu, Han Liu, Yifan Zhu, Xuekai Xiong, Xi Chen, Changmin Hu, Huanchun Chen, Yingyu Chen, Aizhen Guo

**Affiliations:** 1The National Key Laboratory of Agricultural Microbiology, Wuhan 430070, China; wangjr0317@163.com (J.W.); xiaojie.zhu@Murdoch.edu.au (X.Z.); pengyongchong@webmail.hzau.edu.cn (Y.P.); 553238808@webmail.hzau.edu.cn (T.Z.); liuhan18415@163.com (H.L.); avander1@163.com (Y.Z.); xiongxuekai0713@gmail.com (X.X.); chenxi@mail.hzau.edu.cn (X.C.); hcm@mail.hzau.edu.cn (C.H.); chenhch@mail.hzau.edu.cn (H.C.); 2Key Laboratory of Pig Molecular Quantitative Genetics of Anhui Academy of Agricultural Sciences, Institute of Animal Husbandry and Veterinary Medicine, Anhui Academy of Agricultural Sciences, Hefei 230031, China; 3College of Veterinary Medicine, Huazhong Agricultural University, Wuhan 430070, China; 4College of Veterinary Medicine, Murdoch University, Murdoch 6160, Australia; 5National Animal Tuberculosis Para-Reference Laboratory (Wuhan) of Ministry of Agriculture and Rural Affairs, Huazhong Agricultural University, Wuhan 430070, China; 6Hubei International Scientific and Technological Cooperation Base of Veterinary Epidemiology, Huazhong Agricultural University, Wuhan 430070, China

**Keywords:** *Mycobacterium tuberculosis*, YrbE3A, macrophages, proinflammatory cytokines, signaling pathways

## Abstract

*Mycobacterium tuberculosis* is considered a successful pathogen with multiple strategies to undermine host immunity. The YrbE3A is encoded by Rv1964 within the RD15 region present in the genome of *Mtb*, but missing in *M. bovis*, *M. bovis* BCG (Pasteur) strain, and *M. smegmatis* (Ms). However, little is known about its function. In this study, the YrbE3A gene was cloned into pMV261 and expressed in Ms and BCG, while the strains with the vector served as the controls. The YrbE3A was expressed on the mycobacterial membrane, and the purified protein could stimulate RAW264.7 cells to produce IL-6. Furthermore, the effect of the recombinant strains on cytokine secretion by RAW264.7 was confirmed, which varied with the host strains. Ms_YrbE3A increased significantly higher levels of TNF-α and IL-6 than did Ms_vec, while BCG_YrbE3A enhanced higher TNF-α than BCG_vec. The pathways associated with NF-κB p65 and MAPK p38/JNK, other than Erk1/2, regulated this process. In addition, mice were infected with Ms_YrbE3A and Ms-vec and were kinetically examined. Compared to Ms-vec, Ms_YrbE3A induced more serious inflammatory damage, higher levels of TNF-α and IL-6, higher numbers of lymphocytes, neutrophils, and monocytes in a time-dependent way, but lower lung bacterial load in lung. These findings may contribute to a better understanding of Mtb pathogenesis.

## 1. Introduction

In 2015, the World Health Organization initiated a campaign to end tuberculosis (TB), with the hope of globally ending TB by 2035 [1]. However, in 2018, TB remains the leading cause of death from a single infectious agent, with an estimated 10 million new cases and 1.49 million new deaths due to TB [2]. Pathogens such as *Mtb* have evolved numerous virulent effector proteins to disturb the host immune response by targeting key components of the innate immune system [3,4]. Consequently, it is important to identify the virulence-related molecules of *Mtb* (*Mycobacterium tuberculosis*) to better understand its pathogenesis [5,6]. 

The region of differences (RD) is defined as the number of fragments missing from the BCG (*Bacillus Calmette–Guérin*) vaccine compared by comparative genomics with the virulent *M. bovis (Mycobacterium bovis*), and collectively the *Mtb* genomes and the protein encoded by this region were termed as the RD region protein [7]. Faksri reported that the RD region protein is important for the development of new diagnostic methods for tuberculosis, whereas the RD region gene may be associated with the virulence of *M. bovis* and *Mtb* [8,9]. Thus, the study of the function of RD region proteins is critical for understanding the pathogenesis of tuberculosis. The YrbE3A protein encoded by the Rv1964 gene, which belongs to the RD15 region, is present in the genome of *Mtb* but missing in *M. bovis*, *M. bovis* BCG (Pasteur) strain, and *M. smegmatis* (Ms). To date, YrbE3A is an unknown and conserved hypothetical integral membrane protein as a part of the Mce operon and a member of the YrbE family [10,11]. Despite extensive research in mice demonstrating the importance of these operons in infection outcomes, the physiological function of the YrbE protein remains unknown [11,12]. Therefore, we investigated the role of YrbE3A through the construction of recombinant *Ms* and BCG strains expressing YrbE3A in vitro and in vivo in the present study. Our results indicated that the YrbE3A is a stimulator of the proinflammatory cytokines TNF-α and IL-6.

## 2. Materials and Methods

### 2.1. Bacteria and Antibodies

The *M. smegmatis* mc^2^155 strain was kindly provided by Professor Luiz Bermudez from Oregon State University (Corvallis, OR, USA). *M. smegmatis* mc^2^155, *M. bovis*, *M. bovis* BCG (Pasteur) strain, and recombinant strains were cultured in Middlebrook 7H9 broth with or without kanamycin containing 10% oleic acid–albumin–dextrose–catalase (OADC) and 0.05% Tween-80, or in Middlebrook 7H11 agar plates containing 10% OADC [13].

The anti-β-actin antibody (A2228) was purchased from Sigma-Aldrich (Shanghai, China). The anti-GAPDH antibody (E021010) was purchased from EarthOX Company (San Francisco, CA, USA).. Phospho-SAPK/JNK (Thr183/Tyr185) (81E11) Rabbit mAb (#4668), JNK2 (56G8) Rabbit mAb (#9258), anti-NF-κB subunit p65 (L8F6) Mouse mAb (#6956), Phospho-NF-κB p65 (Ser536) (93H1) Rabbit mAb (#3033), Phospho-p38 MAPK (Thr180/Tyr182) Antibody(#9211), p38 MAPK Antibody (#9212), p44/42 MAPK (Erk1/2) (137F5) Rabbit mAb (#4695), and Phospho-p44/42 MAPK (Erk1/2) (Thr202/Tyr204) (D13.14.4E) XP^®^ Rabbit mAb(#4370) antibodies were obtained from Cell Signaling Technology (CST, Danvers, MA, USA).

### 2.2. Mice and Mammalian Cell Lines

Forty-five each of specific pathogen free male and female C57BL/6 mice (6 to 8 weeks old) were obtained from Vital River Company (Beijing, China). Mice were bred and raised in individual cages at the Laboratory Animal Center of Huazhong Agricultural University. All experimental procedures were approved(7 Mar 2017) by the Ethical Committee of Huazhong Agricultural University (Permit number: HZAUMO-2017-037).

RAW264.7 cells were grown in DMEM containing 10% (*v/v*) fetal bovine serum (FBS), 100 mg/mL streptomycin, and 100 U/mL penicillin at 37 °C and 5% CO_2_ [14,15].

### 2.3. Construction of Recombinant Strains and YrbE3A Protein Expression

The YrbE3A gene was amplified from *Mtb* H37Rv genomic DNA using gene-specific primers (Table 1). The digested YrbE3A PCR product was then cloned into the pMV261 vector to generate the recombinant plasmid pMV261-YrbE3A, which was electroporated into wild-type *Ms* mc^2^155 and BCG to generate the Ms_YrbE3A and BCG_YrbE3A strains, respectively. The empty vector pMV261 was also electroporated into *Ms* mc^2^155 and BCG to obtain the *Ms*-vec and BCG_vec strains, respectively, which were used as controls in the following experiments. In order to describe the morphology of the four strains, the colony circularity and diameter of four recombinant strains were measured by ImageJ software according to the instructions [16]. Ten independent colonies were analyzed for each strain. According to the instructions of ImageJ, the circularity was calculated as 4π × area/perimeter^2^. A value of 1.0 indicates a perfect circle. As the value approaches 0.0, it indicates an increasingly elongated shape.

Recombinant Ms_YrbE3A and Ms_vec constructs were grown, and YrbE3A expression was induced in a 45 °C water bath for 1 h, lysed by ultrasonication, and purified by Ni-NTA agarose chromatography (Qiagen, Chatsworth, CA, USA). The protein was then filtered using a 0.22-μm filter. After the above-mentioned treatment, the recombinant protein was stored at −80 °C [17]. 

The YrbE3A protein was encoded by Rv1964, which was approximately 798 bp in size, and the recombinant YrbE3A protein was approximately 32 kDa. The expression of this protein was checked by 12% SDS-PAGE and Western blot assay, as described below. 

### 2.4. Macrophage Infection and CFU Counting

RAW264.7 cells were inoculated into 6-well plates with a density of 5 × 10^5^ to 1 × 10^6^ cells/well and cultured for 12 h before infection. Then, the cells were infected with the strains of Ms_YrbE3A, Ms_Vec, BCG_YrbE3A, and BCG_Vec at an MOI of 10. After two hours, the infected cells were washed three times with phosphate-buffered saline (PBS), and gentamicin (20 mg/mL) was added to kill the bacteria outside the macrophages. For cytokine testing, culture supernatants were collected from infected cells at 0, 2, 4, 8, and 24 h after infection. Cytokines were detected using commercial ELISA kits (New Bioscience, Shanghai, China) according to standard procedures. For bacterial survival within RAW264.7 macrophages, infected cells were lysed in 0.025% (*v/v*) SDS after being washed three times with PBS and 10-fold serially diluted from 100 µL lysate to 900 µL PBS. Then, 100 µL of each dilution were plated on 7H11 agar plates containing 10% glycerol. After a 3-day incubation period, the colonies on each plate were counted. The survival rate was calculated relative to control values [18]. Lung tissue was homogenized in 1 mL of 6% NaOH. Each homogenate was diluted and plated on 7H10 agar plates. Colonies were scored after 7–14 days of incubation at 37 °C.

### 2.5. ELISA and Quantitative RT-PCR Analysis to Detect Cytokine Expression

RAW264.7 cells were infected with Ms_YrbE3A, Ms_Vec, BCG_YrbE3A, and BCG_Vecc at an MOI of 10. After 2, 4, 8, and 24 h of infection, total cellular RNA was extracted from the infected cells using the TRIzol reagent (Invitrogen, Carlsbad, CA, USA) and then was used in a reverse transcription reaction using an RT Kit (Vazyme, Nanjing, China). The quantitative PCR reaction was performed using an Applied Biosystems Viia 7 Fast Real-Time qPCR System (Applied Biosystems, Carlsbad, CA, USA) and SYBR Green Master Mix (Vazyme, Nanjing, China). For each sample, relative mRNA levels were evaluated after being normalized to that of β-actin or GAPDH. The primers used in this step are listed in Table 1.

The culture supernatants were collected from infected macrophages, and cytokine production, including IL-6, TNF-α, and IL-1β, was detected with commercial ELISA Kits (New Bioscience, Shenzhen, China) following the manufacturer’s instructions. 

To check the effect of recombinant YrbE3A protein on cytokine expression, RAW264.7 cells were treated with LPS (1 μg/mL) (Sigma, Shanghai, China) or varying concentrations (0, 2, 5, or 10 μg/mL) of purified recombinant YrbE3A protein from Ms_YrbE3A for 18 h. Then, IL-6 levels in the supernatant of the cell culture were detected with the above-mentioned ELISA.

All assays were performed in triplicate, and three independent experiments were conducted. Data are expressed as mean ± SEM [15].

### 2.6. Localization of the YrbE3A Protein 

First, 200 mL of the recombinant constructs, Ms_ YrbE3A and Ms _Vec, were grown to an OD_600_ of ~0.8 and subjected to cellular fractionation. Next, the cells were harvested and resuspended in 2 mL lysis buffer (0.1 MPBS, 1 mM PMSF) with 200 uL glass beads and then beat for 60 s. Then, the soluble protein (cytoplasmic protein) was collected by centrifugation at 15,000× *g* for 1 min at 4 °C, and the precipitate was performed with 1 mL 1% (*w/v*) SDS and bead-beating for 60 s. The supernatant was collected as the cell membrane fraction by centrifuge at 15,000× *g* for 1 min. The cytoplasmic protein and cell membrane fraction were stored at −20 °C for Western blot analysis. 

### 2.7. Western Blot Assays of Molecules Critical to Related Signal Pathways

Western blot analysis was performed to confirm the YrbE3A-induced phosphorylation of signal transduction molecules. To extract the cytosolic protein, after stimulation with the strains Ms_YrbE3A, Ms_Vec, BCG_YrbE3A, and BCG_Vec at an MOI of 10 for various times from 0 to 24 h, RAW264.7 cells were lysed in cell lysis buffer and washed twice with ice-cold PBS. Then, the cellular lysates were incubated for 10 min on ice and centrifuged at 12,000× *g* for 10 min at 4 °C. The protein concentration of the lysates was measured using a BCA Protein Assay Kit (Tian Gen, Beijing, China). An equal amount of cell lysates or cellular membrane proteins was then separated by SDS-PAGE and transferred to PVDF membrane. Membranes were blocked in 5% non-fat dry milk with TBST and incubated overnight with the primary antibodies against phosphorylated forms of p65, JNK, ERK1/2, and p38, respectively. β-actin served as the internal control. Then, the membranes were incubated with horseradish peroxidase (HRP)-conjugated goat anti-rabbit (1:1000) and anti-mouse immunoglobulin (Ig)G (1:1000) (Southern Biotech, Birmingham, AL, USA) at room temperature for 1 h, and developed using Immobilon Western Chemiluminescent HRP Substrate (DNR, Jerusalem, Israel), according to the manufacturer’s instructions [19].

### 2.8. Effect of Signaling Pharmacological Inhibition on Cytokine Production

All pharmacological inhibitors were purchased from Sigma-Aldrich (Shanghai, China), reconstituted in sterile DMSO (Sigma-Aldrich, Shanghai, China), and utilized at varying concentrations. The pharmacological inhibitors included the ERK signaling inhibitor U-0126 (10 mM), the NF-κB signaling inhibitor PDTC (20 mM), and the JNK signaling inhibitor SP600125 (20 mM) (Cayman Chemical, Ann Arbor, MI, USA). DMSO, at a concentration of 0.1%, was used as the vehicle control. In the inhibitory experiments, cells were treated with a given inhibitor for 30 min before treatment with Ms_YrbE3A and BCG_YrbE3A. Culture supernatants were then harvested at 8 h after stimulation with Ms_YrbE3A and BCG_YrbE3A and assayed for cytokine concentrations [15].

### 2.9. Mouse Infection

The 90 C57BL/6 mice (45 female and 45 male) were divided into 3 groups of 30 each for the respective infection with Ms_YrbE3A, Ms_Vec, or PBST (PBS with 0.05% Tween 80). Briefly, the bacilli suspended in PBST were intratracheally delivered to corresponding groups of animals at 2 × 10^6^ CFU per mouse in 25 µL PBST, while the same amount of PBST was administered into the control mice. On days 0, 2, 4, 8, 16, and 21 postinfection (PI), 5 mice at each timepoint were euthanized, and the lungs were collected for examination of histopathology, bacterial load, and inflammatory cytokines. 

### 2.10. Histopathological and Immunohistochemical Examination

For histopathology examination, lung samples were fixed in 10% neutral buffered formalin, which was made with PBS (pH 7.4, 0.1 M), immediately after collection for histopathological and immunohistochemical examination. Paraffin-embedded sections (4 μm thick) were cut, and conventional hematoxylin and eosin (H&E) staining was performed for histopathological observation [17]. These sections were evaluated in a blinded fashion by a pathologist. For quantitative analysis of inflammatory cells lymphocytes, neutrophils, and monocytes, one slide for each lung was randomly selected, and 10 fields of each slide were evaluated for cell counting using light microscopy. The total cell amount for each group was then compared.

The existence of IL-6 and TNF-α in the lungs was determined after immunohistochemical staining with the IL-6 and TNF-α antibodies. One slide for each lung was selected randomly, and 5 fields of each slide were evaluated under light microscopy. A brown positive signal was quantitatively expressed as the integrated optical density (IOD), and the average IOD per field for individual lungs was calculated using Image-Pro Plus 6.0 (IPP6) software (Media Cybernetics, Inc., Bethesda, MD, USA). A comparison of the total IODs between each of the two groups was performed [20].

### 2.11. The Multiplex Immunoassay for Inflammatory Cytokines in Lungs

The surface of mice lung was seared. Then, 20 mg of lung was cut using a sterile scalpel blade and put into a clean, labelled 2 mL microcentrifuge tube containing stainless glass beads and 250 μL PBS. Tissue samples were then crushed using a TissueLyser II (Qiagen, Chatsworth, CA, USA), and then the lung homogenates were subjected to the multiplex immunoassay for inflammatory cytokines by Antgene Biotech (Wuhan, China), which includes TNF-α, IL-1β, IL-6, IFN-γ, MCP-1, IL-1α, IL-12p70, and IL-10.

### 2.12. Statistics

Data in the graphs are expressed as mean ± SEM of triplicate experiments. Statistical testing was conducted using a two-tailed unpaired *t*-test with Welch’s correction and a two-way ANOVA followed by the LSD test. Significant differences are indicated by an asterisk and were determined using GraphPad 5.0 Software (San Diego, CA, USA). The values of **p* < 0.05, ***p* < 0.01, and ****p* < 0.001 were considered statistically significantly different.

## 3. Results

### 3.1. YrbE3A Expression and Growth Characterization of Ms_YrbE3A and BCG_YrbE3A Strains

The correctness of YrbE3A insertion in pMV261-YrbE3A was confirmed by PCR and sequencing (data not shown). The expression of YrbE3A protein was detected by 12% SDS-PAGE with a specific band of about 32 kDa (Appendix A). Further, the membrane location of this protein in Ms_YrbE3A was identified with Western blot assay by using mAb to His-tag (Appendix A). 

To test the effect of YrbE3A on the growth of recombinant strains, the colony morphology was observed. Generally speaking, the colonies of the YrbE3A expressing strains Ms_YrbE3A and BCG_YrbE3A were larger than their control strains Ms_Vec and BCG_vec (Figure 1A). By using ImageJ to measure the colonies, the circularity of the Ms_YrbE3A and Ms_vec colonies was determined to be 0.36 ± 0.05 and 0.71 ± 0.01, respectively, showing a significant difference (*p* < 0.001). The average colony diameter of Ms_YrbE3A was significantly larger than that of Ms_vec (*p* < 0.001) (Figure 1B). Similarly, the colony diameter and circularity of BCG_YrbE3A were larger than those of BCG_Vec (*p* < 0.05) (Figure 1A and 1B). 

In addition, the growth curves of these four strains were detected and compared. No growth difference was detected between the Ms_YrbE3A and Ms_Vec strains at most time points, except at 12 h and 24 h, when Ms_YrbE3A grew significantly faster than Ms_Vec (*p* < 0.001) when cultured in Middlebrook 7H9 broth for 144 h (Figure 1C left panel). However, the BCG_YrbE3A strain had a significantly more rapid growth compared with the BCG_Vec strain from 24 d to 41 d at the stationary phase (Figure 1C right panel). 

### 3.2. YrbE3A Reduced Mycobacterial Intracellular Survival and Induced Cytokine Production

To determine whether YrbE3A is essential for mycobacterial virulence, we analyzed the intracellular survival of the recombinant bacteria in macrophages. To achieve this, we infected RAW264.7 cells with Ms_YrbE3A, BCG_YrbE3A, Ms_Vec, and BCG_YrbE3A (MOI = 10:1). The four strains were dispersed into single-cell suspensions, and CFUs were counted at 2, 4, 8, and 24 h postinfection (hpi). Both Ms_YrbE3A and BCG_YrbE3A expressing YrbE3A showed significantly lower bacillary counts in RAW264.7 cells at 2, 4, and 8 hpi (Figure 2A). These results suggested that the expression of YrbE3A had the ability to reduce the intracellular survival of Ms_YrbE3A and BCG_YrbE3A in macrophages.

To check the stimulating effect of recombinant YrbE3A protein on cytokine secretion, IL-6 concentrations were tested in the supernatants of RAW264.7 cells treated with purified YrbE3A protein, as well as LPS as the positive control. The results showed that YrbE3A protein at 2 μg/mL could significantly induce IL-6 production compared with the control, and 5 μg/mL YrbE3A stimulated the highest expression of IL-6, higher than 1 μg/mL of LPS (Figure 2B). Then, the effect of recombinant bacteria Ms_YrbE3A and BCG_YrbE3A expressing YrbE3A protein on the production of the pro-inflammatory cytokines TNF-α, IL-1β, and IL-6 was checked. The RAW264.7 cells were infected with Ms_YrbE3A/Ms_Vec and BCG_YrbE3A/BCG_Vec, and the levels of the cytokines were compared. Generally speaking, TNF-α and IL-6 production showed a higher level than IL-1β. Compared to the control strain, Ms_YrbE3A promoted RAW264.7 cells to produce significantly higher concentrations of TNF-α and IL-6 between 4 and 24 hpi, but not IL-1β (Figure 2C, Appendix A). At the mRNA level, only at 24 hpi was the expression of all three cytokines in Ms_YrbE3A infected cells significantly higher than Ms_Vec infected cells (*p* < 0.001) (Appendix A). 

However, compared to Ms_YrbE3A, BCG_yrbE3A showed a weaker ability to stimulate cytokine production (Figure 2D, Appendix A). Among three cytokines tested, only TNF-α expression was significantly up-regulated. At the protein level, only at 24 hpi did BCG_yrbE3A infection significantly induce a higher level of TNF-α production than the BCG_Vec strain (*p* < 0.01) (Figure 2D).

### 3.3. MAPK-JNK/p38 and NF-κB p65 Signaling Were Involved in the YrbE3A-Mediated Induction of Cytokines

To further elucidate the mechanisms through which the YrbE3A protein induced cytokine secretion, we tested molecules critical to the key signal transduction pathways in YrbE3A-stimulated RAW264.7 cells. We measured the phosphorylation of NF-κB p65 and MAPK in the RAW264.7 cells induced by four strains at 0, 2, 4, 8, and 24 hpi by Western blot assay (Figure 3A,B). Our results revealed that the phosphorylation of MAPK-p38 (*p*-p38) and NF-κB p65 (p-p65) was significantly enhanced by Ms_YrbE3A infection at early stages, but phosphorylation of MAPK-JNK (p-JNK) occurred at late stages, showing a time-independent manner. However, the BCG_YrbE3A infection stimulated a significant increase in p-p65 at a late stage (8 and 24 hpi) and p-p38 at 8 hpi, but not in p-JNK. Both strains had no accelerated effect on the activation of Erk1/2.

To confirm the above results, RAW264.7 cells were pretreated with the NF-κB p65 specific inhibitor SC-514, the ERK specific inhibitor U-0126, and the JNK specific inhibitor SP600125 for 30 min before Ms_YrbE3A and BCG_YrbE3A infection [21,22,23]. Then, we tested the effect of the above inhibitors on the production of TNF-α and IL-6 in RAW264.7 at 8 hpi. As presented in Figure 3C, the cells treated with SC-514 and SP600125 showed a significant inhibition of TNF-α and IL-6 expression in Ms_YrbE3A infected cells and TNF-α in BCG_YrbE3A infected cells (*p* < 0.001). However, U-0126 did not display any significant inhibitory effect on both cytokines in the RAW264.7 cells infected by these two strains. These findings are in agreement with the above results about the expression of phosphorylated signal molecules. Overall, the NF-κB p65 and MAPK-JNK/p38 are involved in YrbE3A mediated production of TNF-α and IL-6 for Ms_YrbE3A, and NF-κB p65 and MAPK-p38 are involved in YrbE3A mediated production of TNF-α for BCG_YrbE3A.

### 3.4. YrbE3A Exacerbated Histopathology during Ms_YrbE3A Infection

The role of YrbE3A in the induction of a pro-inflammatory response was then assessed in C57BL/6 mice using a dose of 2.0 × 10^6^ CFU. We performed an in-depth analysis of disease parameters at 0, 2, 4, 8, 16, and 21 days postinfection. Overall, the severity of lung damage increased with time. In addition, lung damage caused by Ms_YrbE3A was more serious than the Ms-vec. From 2 days after infection, the histopathological change in the lungs became obvious, as shown by the increase in the alveolar wall, infiltration of immune cells, fusion of the alveolar cavities, and finally, the collapse of the lung structure (Figure 4A, Appendix A). 

Regarding the infiltration of immune cells in the lungs, three types of cells, including lymphocytes, neutrophils, and monocytes, were quantified as cell numbers per field. Basically, for the Ms-vec strain infected mice, only the number of neutrophils at 21 days increased significantly (*p* < 0.01), while the numbers of lymphocytes and monocytes at all time points and neutrophils at the time points between 2 to 16 days did not show a significant difference from each other, although they experienced some fluctuation (*p* > 0.05). However, Ms_YrbE3A induced rapid increase of lung lymphocytes and reached the peak at 2 days PI. Then, the number of lung lymphocytes gradually decreased but maintained high levels at 4 and 8 days PI, and dropped to the basic level of day 0 at 16 and 21 days PI. On the contrary, the number of neutrophils increased in the late stages of infection, reaching the peak at day 16 PI and then decreased but still kept a significantly higher level at day 21 PI. Among the three types of immune cells, monocytes had the smallest increase (Figure 4B).

To further investigate how YrbE3A influenced the mouse pro-inflammatory response during infection in vivo, we compared the cytokine production in Ms_YrbE3A-infected mice with Ms_Vec and saline control groups. Ms_YrbE3A infection resulted in the highest concentrations of pro-inflammatory cytokines, including TNF-α, IL-1β, IL-6, MCP-1, IL-1α, IL-12p70, and IFN-γ, in the lungs at 2 d PI and later decreasing rapidly from 4 to 21 d PI. In contrast, the anti-inflammatory IL-10 was observed to decline at 2 d PI, later increasing gradually from 4 to 21 d PI (Figure 5A, Appendix A). In addition, immunohistological assay of lung tissues demonstrated that IL-6 was significantly increased at 2 and 4d PI and decreased to a normal level after 8 d. TNF-α was significantly increased from 4 to 21 days compared with Ms_Vec-infected mice (Figure 5C,D, Appendix A).

Meanwhile, the lung bacterial load was tested. Both strains were recovered between day 0 and 8 PI. After that time, no bacteria were recovered. Ms_Vec experienced a significant growth in mice at day 2 PI (*p* < 0.001) and then rapidly decreased at day 4, but was still higher than the basic level of day 0 (*p* < 0.05), and thereafter kept a decreasing trend to an undetectable level. However, Ms_YrbE3A did not grow in vivo and decreased continuously to an undetectable level (Figure 5B). 

## 4. Discussion

Host innate immune responses, particularly those involving inflammation, are critical at the early stages of infection in order to effectively clear pathogens [24,25]. Many cytokines regulated by MAPK and NF-κB signaling pathways have been implicated in host–mycobacterial interactions. There are increasing numbers of examples of *Mtb* that have evolved numerous virulent effector proteins to disturb the host immune response by targeting key components of the innate immune system. By way of illustration, PPE17 and PPE65 bind to TLR2 and generate a pro-inflammatory response [26,27]. In this study, we reveal a novel *Mtb* virulence-related factor, Yrb3EA, which regulates the host inflammatory response both in vivo and in vitro through MAPK/JNK and NF-κB signaling pathways, and concurrent with the increased expression of inflammatory cytokines such as TNF and IL-6. 

### 4.1. The Character of Yrb3EA Protein

The Yrb3EA protein in *Mtb* is localized in the inner membrane with the help of five TM segments that have been predicted using TMHMM 2.0 [28]. This protein is the homologue to the MlaE protein in *E. coli*, which is a component of the Mla pathway and an ABC transport system that functions to maintain the asymmetry of the outer membrane. It has been reported that bacterial ABC transporters are necessary for the growth and survival of the bacteria in their ecological niches [3,19]. In addition, there is increasing evidence that these transport systems play direct and/or indirect roles in the virulence of bacteria [29]. However, it is still unclear how YrbE3A is involved in interactions with the host during the *Mtb* infection. In this study, we constructed YrbE3A overexpression nonpathogenic fast-growing Ms and BCG to investigate its function. We optimized conditions of YrbE3A protein expression by using different IPTG and induction times and transformed recombinant vectors into different competent cells, including rosetta and C43(DE3), for the efficient expression of this hydrophobin (data not shown). However, since the Yrb3EA protein in *Mtb* is localized in the inner membrane, we were unable to obtain enough soluble protein for identification of Yrb3EA function in vitro and in vivo. 

### 4.2. How Yrb3EA Affects Cytokine Expression in Vitro and in Vivo

In this work, we found that the YrbE3A protein could significantly activate the inflammatory response in RAW264.7 cells and mice. Ms_YrbE3A induced IL-6 and TNF-α expression in RAW264.7 at various time points starting from 2 h after infection. In the mouse experiment, the induction of IL-6 and TNF-α was confirmed. Moreover, several other cytokines, namely IL-12, IFN-γ, MCP-1, and IL-1α, which are critical to innate and acquired immunity, were rapidly up-regulated at day 2 PI after infection. This effect of YrbE3A protein in vivo should depend on the complicated neuroendocrine immune network of mice. To support this assertion, besides for the cytokines, we also showed that YrbE3A kinetically activates differentiation and proliferation of different immune cell lymphocytes, monocytes, and neutrophils in mouse lungs. In early stages of infection, lymphocytes increased significantly, while in late stages, neutrophils rose greatly [30,31]. In addition, this was consistent with another ABC transporter protein, Rv1273c, which has been previously shown to act as a strong immune modulator and is a potent stimulator of pro-inflammatory cytokines [19]. Besides, these results agree with the previous studies that mycobacteria-infected macrophages or monocytes are shown to secrete both pro-inflammatory cytokines, including IL-1, IL-6, IL-12, and TNF-α, and anti-inflammatory cytokines, as well as IL-4 and IL-10 [32,33]. 

However, compared to Ms_YrbE3A, BCG_YrbE3A only significantly stimulated production of TNF-α among the three tested cytokines (TNF-α, IL-6, and IL-1β) in macrophages. In agreement with in vitro findings, immunohistochemical staining of lung sections showed that expression of TNF-α, but not IL-6, became obvious at day 4 PI and maintained high levels at days 8 and 16 PI. Therefore, TNF-α is the cytokine YrbE3A mostly significantly induced. As previously reported, TNF-α is involved in the host defense against mycobacterial infection and can increase the survival rate and induce granuloma formation [33,34]. This lack of in vivo confirmation of BCG_YrbE3A is a limitation of this study and needs to be investigated in the future.

### 4.3. How the MAPK and NF-κB Signaling Pathways Are Involved in Cytokine Expression

When the host responds to pathogens, the NF-κB and MAPK pathways are essential for the inducible expression of multiple inflammatory genes. A previous study illustrated that Mce3E, another homologous protein of *Mtb,* could suppress both the JNK and ERK signaling pathways [25]. In order to understand the mechanism of YrbE3A-induced inflammation, we performed a Western blot assay of NF-κB, MAPK, and ERK pathways and their inhibitory experiments. We demonstrated that YrbE3A promoted the production of cytokines by targeting both NF-κB and MAPK pathways by directly phosphorylating p-p65, p-JNK, and p-p38. In agreement, p65 and JNK pathway inhibitors suppressed TNF-α expression in Ms_YrbE3A infected cells. TNF-α was previously demonstrated to be regulated by NF-κB and MAPK pathways [35,36]. Therefore, YrbE3A regulated cytokines by NF-κB p65 and MAPK-JNK signaling. 

### 4.4. The Overexpression of YrbE3A Inhibits the Survival of Mycobacteria in Vitro and in Vivo 

Previous studies have reported that TNF-α can not only induce apoptosis of macrophages involving apoptosis signal-regulated kinase 1 and mitogen-activated protein kinase (MAPK) p38, but also can activate macrophages to produce NO via NOS2 using L-arginine as the substrate, thereby inhibiting and/or killing *M. tuberculosis* in the human system [37,38,39]. Moreover, a number of studies have suggested that TNF can induce autophagy, which also can promote the fusion of the MTB phagosome with autophagosomes and facilitates subsequent clearance of the bacilli in autophagolysosomes [40,41]. Since the transcriptional and translational expression of TNF-α increased during Ms_YrbE3A and BCG_YrbE3A infection, we conducted experiments to understand whether YrbE3A can also affect the survival rate of mycobacteria. Although the expression of YrbE3A protein did not impede mycobacterial growth under medium conditions, this study demonstrated it to be associated with the restriction of mycobacterial survival in both macrophages and mice. In this study, we noticed that two hours after macrophage infection, fewer Ms_YrbE3A strains survived in the macrophages compared with the Ms_Vec strain; moreover, cytokine expression was markedly increased in the cells infected with the Ms_YrbE3A and BCG_YrbE3A strains. In the mouse experiment, the lung load of Ms_Vec strain significantly increased at days 2 and 4 PI, and then later decreased. However, the lung load of Ms_YrbE3A decreased continuously in infected strains. The results showed that YrbE3A enhances the bactericidal activity of macrophages by increasing the expression of TNF-α. Yao et al. have reported that Rv2346c can enhance the survival of mycobacteria in macrophages by inhibiting TNF-α and IL-6 production via the p38/miRNA/NF-κB pathway [38]. Deletion of PtpA in BCG also increased expression of TNF and IL-1β and reduced the survival of mycobacteria in U937 cells [17]. Based on the effect of YrbE3A protein on cytokine production and proliferation of immune cells, the restriction on mycobacterial survival by YrbE3A expression could be realized by the enhanced inflammatory response and likely acquired immunity. 

In conclusion, our results show that the YrbE3A protein can stimulate production of pro-inflammatory cytokines such as TNF-α through NF-κB and JNK MAPK signaling pathways and inflammatory response both in vivo and in vitro to restrict mycobacterial replication and survival. Therefore, we revealed a novel virulence-related factor of Mtb. This finding may contribute to a better understanding of Mtb pathogenesis and provide a potential drug target for TB treatment.

## Figures and Tables

**Figure 1 microorganisms-08-00584-f001:**
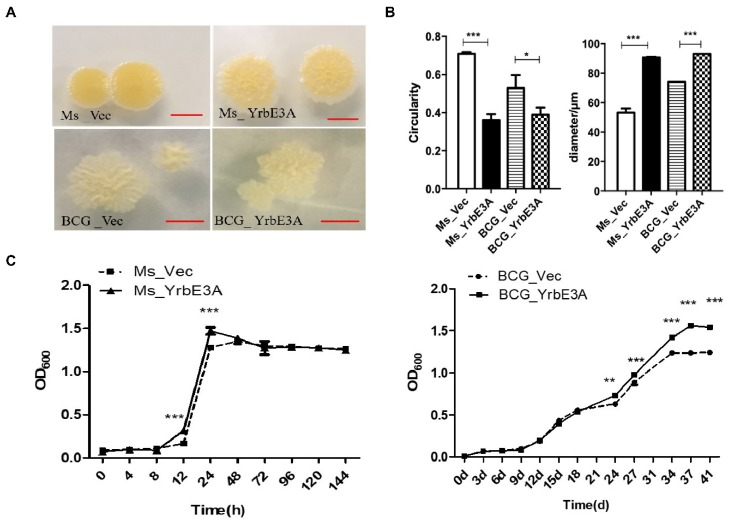
The effect of YrbE3A expression on colony morphology and bacterial growth of the recombinant strains. (**A**) The morphology of Ms_YrbE3A and Ms_Vector (upper panel) and BCG_YrbE3A and BCG_Vector (lower panel), scar bar = 5 mm. (**B**) The colony circularity and diameter of four recombinant strains were measured by ImageJ software. Ten independent colonies were analyzed for each strain, and data were expressed as the mean ± SD from three separate experiments. **p* < 0.05, ****p* < 0.001 (two-tailed unpaired t-test). (**C**) Bacterial growth curves were shown by OD_600_ values of the broth cultures of Ms_YrbE3A and Ms_Vector (left panel) and BCG_YrbE3A and BCG_Vector (right panel). Significant differences (**p* < 0.05, ***p* < 0.01, ****p* < 0.001) among each treatment were analyzed by a two-way ANOVA followed by the LSD test. Each sample was repeated in triplicate.

**Figure 2 microorganisms-08-00584-f002:**
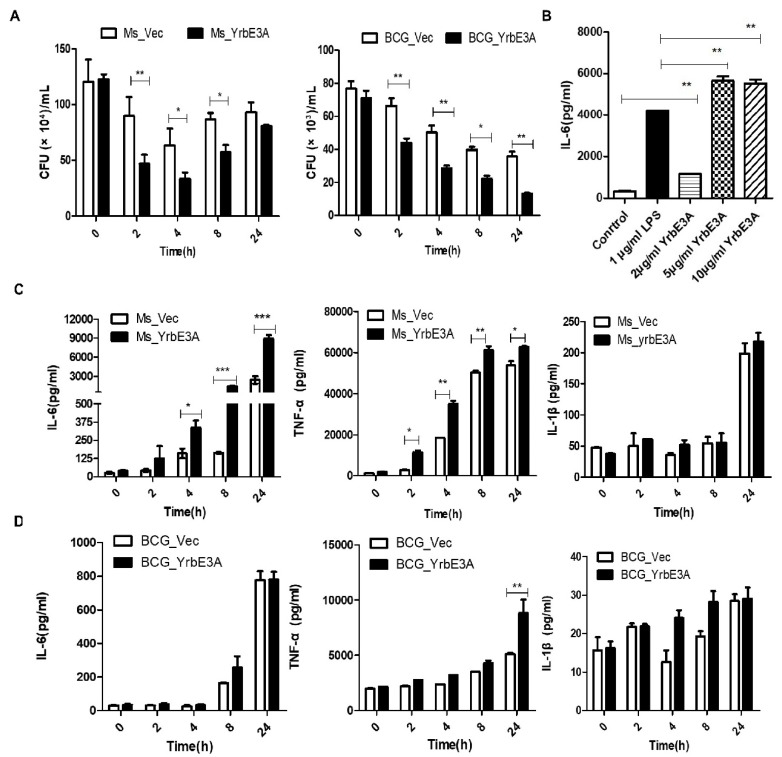
Mycobacterial survival and detection of cytokine expression of Ms_YrbE3A, Ms_Vec, BCG_YrbE3A, and BCG_Vec in infected RAW264.7 cells. (**A**) Intracellular mycobacteria of Ms_YrbE3A, Ms_Vec, BCG_YrbE3A, and BCG_Vec were counted at 0, 2, 4, 8, and 24 h postinfection. (**B**) Detection of IL-6 expression by RAW264.7 specific to different concentrations of purified YrbE3A. ELISA of IL-6 in the supernatants of RAW264.7 cells treated with LPS (1 μg/mL) alone or with varying concentrations of recombinant YrbE3A protein (0, 2, 5, or 10 μg/mL). RAW264.7 without LPS and YrbE3A protein served as control. Data are expressed as the mean ± SD from three separate experiments. **p* < 0.05, ***p* < 0.01 (two-tailed unpaired t-test). (**C,D**) ELISA was used to detect the production of TNF-α, IL-6, and IL-1β by RAW264.7 infected by Ms_YrbE3A and Ms_Vec (**C**) and BCG_YrbE3A and BCG_Vec (**D**). Significant differences (**p* < 0.05, ***p* < 0.01, ****p* < 0.001) among each treatment were analyzed by a two-way ANOVA followed by the LSD test. Each sample was repeated in triplicate.

**Figure 3 microorganisms-08-00584-f003:**
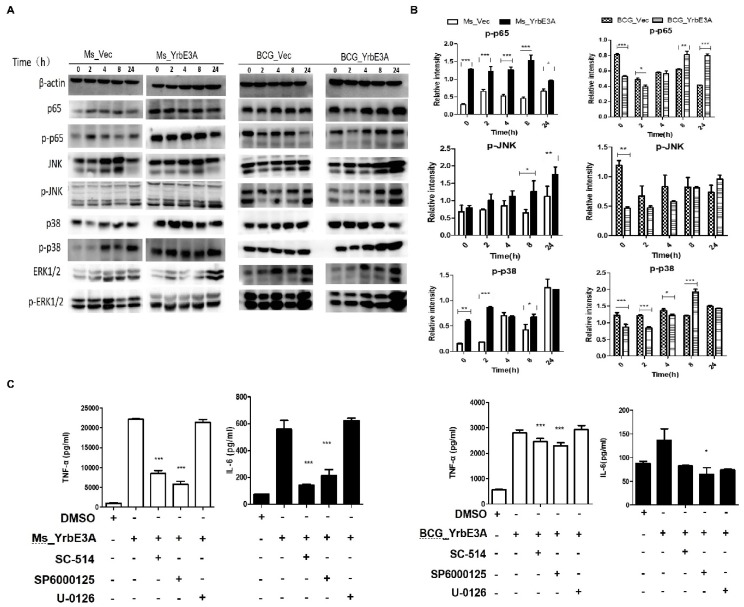
Signal transduction pathways involved in Ms_YrbE3A and BCG_YrbE3A-stimulated pro-inflammatory response in RAW264.7 macrophages. (**A**) Immunoblot analysis of phosphorylated p65, Jnk, p38, and Erk1/2 and total β-actin in RAW264.7 cells. (**B**) Densitometry quantification of immunoblot analysis results of phosphorylated p65, Jnk, and p38 presented relative to those of β-actin. The cells were infected with Ms_YrbE3A/Vec (left) and BCG_YrbE3A/Vec (right) at an MOI of 10 for 0–24 h and sampled at 0, 2, 8, and 24 h postinfection. (**C**) RAW264.7 cells were re-treated with (+) or without (-) DMSO, SC-514, U-0126, and SP600125 for 30 min prior to Ms_YrbE3A (left) and BCG_YrbE3A (right) infection. The culture supernatants were collected at 8 h after infection, and the concentrations of TNF-α and IL-6 were determined. Data are expressed as mean ± SD of the results of three independent experiments. *, **, and *** represent *p*  <  0.05, 0.01, and 0.001, respectively.

**Figure 4 microorganisms-08-00584-f004:**
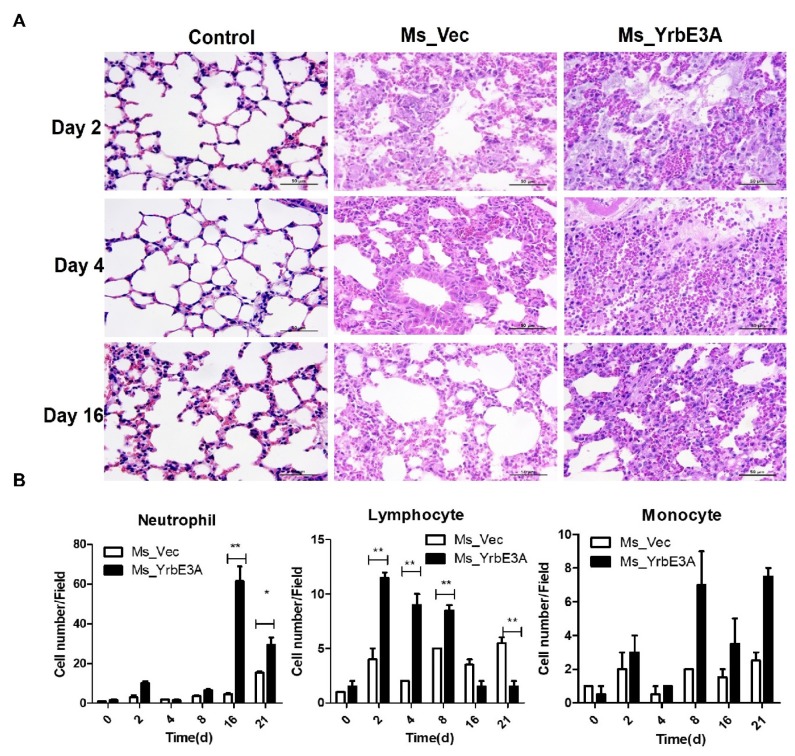
YrbE3A expression promoted lung inflammatory lesions in mice infected with recombinant Ms_YrbE3A and Ms_Vec. (**A**) Mice were infected with Ms_YrbE3A and Ms_Vec at a dose of 2.0 × 10^6^ CFU via intratracheal challenge for different times, euthanized at day 2, 4, and 6 postinfection, and lung tissues were observed after being sectioned and stained with Hematoxylin and eosin (H&E) stain. Scale bar: 50 µm. Data are representative of one experiment with two independent biological replicates (mean ± SEM of *n* = 5 mice per group in (**A**). (**B**) Densitometry quantification of cell numbers per field for neutrophils, lymphocytes, and monocytes **p* < 0.05 and ***p* < 0.01 (two-tailed unpaired *t*-test).

**Figure 5 microorganisms-08-00584-f005:**
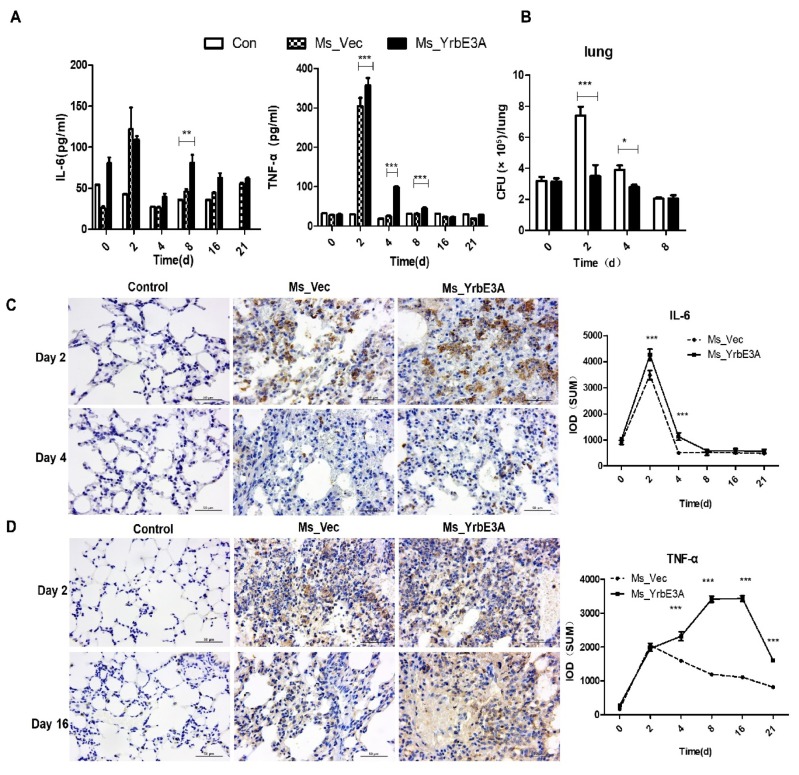
Detection of lung IL-6 and TNF-α and bacterial load in Ms_YrbE3A- and Ms_Vec-infected C57BL/6 mice. (**A**) Multiplex immunoassay of TNF-α and IL-6 in lung cells from C57BL/6 mice infected intratracheally for 0–21 d with 2.0 × 10^6^ CFU of MS_Vec and MS_YrbE3A strains, while a saline control was used as the placebo. (**B**) Lung bacterial load of Ms_YrbE3A and Ms_Vec was detected after 0, 2, 4, and 8 d postinfection. (**C,D**) The detection of the cytokines IL-6 and TNF-α in the lungs of experimental mice by immunohistochemistry (left panel). The brown positive signal produced by immunohistochemistry was quantitatively expressed as integrated optical density (IOD) of IL-6 and TNF-α for the immunohistochemical detection (right panel). Significant difference (**p* < 0.05, ***p* < 0.01, ****p* < 0.001) among treatments was analyzed by a two-way ANOVA followed by the LSD test. Each sample was repeated in triplicate.

**Table 1 microorganisms-08-00584-t001:** Primers used in this study for gene cloning and qRT-PCR.

Genes	Direction	Sequences (5′-3′)
Rv1964	F-BamH I	GCCAAGACAATTGCGGATCCATGGTAATCGTGGCCGACAAG
	R-EcoR V	TGGTGGTGGTGGTGGATATCTCAGGACACCATGAATGGGATG
IL-6	Sense	TGCCTTCTTGGGACTGAT
	Antisense	CTGGCTTTGTCTTTCTTGTT
TNF-α	Sense	CGATGAGGTCAATCTGCCCA
	Antisense	CCAGGTCACTGTCCCAGC
IL-1β	Sense	GCTAGGCTGCTGAGGTTTCTT
	Antisense	TGAAATGCCACCTTTTGACAG
β-actin	Sense	TGCTGTCCCTGTATGCCTCT
	Antisense	GGTCTTTACGGATGTCAACG

Notice: The underscores were the sites for restriction digest.

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
