# Peer review of "Mycobacterium tuberculosis YrbE3A Promotes Host Innate Immune Response by Targeting NF-κB/JNK Signaling"

_microorganisms, 2020, doi:10.3390/microorganisms8040584_

Round 1
Reviewer 1 Report
Investigating how mycobacterial virulence is programmed is critical to understanding how pathogenesis takes place. The authors sought to investigate the role of Mtb YrbE3A by expressing it in the attenuated Tb strain, BCG, and in M. smegmatis. The authors assert that M. smegmatis expressing YrbE3A elicits increased secretion of host immuno-stimulatory factors. The authors speculate that expression of YrbE3A contributes to inflammation during infection with M. tuberculosis. The authors provide data to support several claims, but the manuscript requires significant additional data to more thoroughly support their conclusions.
Major comments:
- Inflammation is associated with TB infection. How important is YrbE3A for eliciting a pro-inflammatory response considering that no significant increase in cytokine production is observed in the recombinant BCG strain?
- The authors assert that cytokine expression is blocked by inhibitors of specific cell signalling modulators, however, the evidence from figure 1 would suggest that this is not necessarily due to YrbE3A. Wouldn't the same trends be observed in the recombinant strains expressing only the empty vector?
Minor comments:
- The figures are so small that they are rather illegible. Also, several of the figures are difficult to decipher what is actually being presented.
- The figure legends must better describe what is being presented int he figures.
- Figure 1a: A loading control is necessary for each of the samples. Also, how do we know whether there is actually a band somewhere on that gel that represents YrbE3A? A tagged fusion could be helpful in order to reveal protein expression on a Western Blot.
- Figure 1b: A quantitation of colony morphology would be helpful. Otherwise, it is difficult to see the differences in the colony morphology as printed. Also, how representative are these colonies of the recombinant strains? The pictures are small and, if printed in black and white, will be completely indiscernible.
- How do you know that yrbE3A is being expressed?
- Is yrbE3A expression must be controleld in figure 2 B-E.
- Line 214: IL-1beta does not appear to be significantly more expressed in Ms_yrbE3A versus Ms_Vec in figure 2b.
- In figure 2c, how different are the relative levels of mRNA expression between each of the two samples?
- How significant is the importance of YrbE3A for eliciting a pro-inflammatory response to the recombinant strain BCG_yrbE3A? The levels are not significantly different for any of the cytokines tested by ELISA except for TNF-alpha at 24h of infection.
- Is purified YrbE3A sufficient to elicit a cytokine response or must it be exposed as a trans-membrane protein?
- Does the expression of YrbE3A reflect the same level and kinetic profile as in TB?
- If growth impacted by higher or lower levels of YrbE3A expression?
- Figure 3c does not seem to illustrate any difference in the expression of the cellular factors assayed. A quantitation of the levels of expression could be helpful in more easily interpreting these results.
- Figure 3d is so small that I can't see it well enough to judge whether the evidence is sufficient to support their claims.
- Figure 4b shows that lymphocytes are initially more highly present than in the sample infected with Ms_Vec. Why does it flip later during infection?
- The red arrows are poorly visible and the phenotype intended to be highlighted is not at all discernible. As such, I can't judge the merit of this figure.
- Figure 5 has panels that are also too small to be reliably judged.
Author Response
Dear reviewer,
Thanks very much for your comments concerning our manuscript entitled “Mycobacterium tuberculosis YrbE3A promotes host innate immune response by targeting NF-κB/JNK signaling” (ID: microorganisms-737544). In according with your advice and comments, we have made extensive modification on the original manuscript and added several extra data as required including:1.western blot assay of the recombinant YrbE3A; 2. the effect of purified YrbE3A at various concentration by using LPS on stimulation of the cytokine TNF-α; 3.the quantification of the colony size of the recombinant strains, etc. Revised portions are marked in red in the revised version of paper. The line number is followed by PDF format. In addition, the manuscript has been edited by the native English-speaking expert. We hope the revised version has been improved to meet with your requirement for publication consideration. The attached is our point-by-point response to your comments.
With best regards,
Aizhen Guo
Investigating how mycobacterial virulence is programmed is critical to understanding how pathogenesis takes place. The authors sought to investigate the role of Mtb YrbE3A by expressing it in the attenuated Tb strain, BCG, and in M. smegmatis. The authors assert that M. smegmatis expressing YrbE3A elicits increased secretion of host immuno-stimulatory factors. The authors speculate that expression of YrbE3A contributes to inflammation during infection with M. tuberculosis. The authors provide data to support several claims, but the manuscript requires significant additional data to more thoroughly support their conclusions.
Major comments:
- Inflammation is associated with TB infection. How important is YrbE3A for eliciting a pro-inflammatory response considering that no significant increase in cytokine production is observed in the recombinant BCG strain?
A: Thanks a lot for your question. Although the recombinant M.smegmatis expressing YrbE3A showed a stronger ability to stimulate production of the three cytokines than BCG_yrbE3A, the latter BCG_yrbE3A induced significantly higher level of TNF-α than BCG_vec at protein level (p<0.01) (Fig.2E).
- The authors assert that cytokine expression is blocked by inhibitors of specific cell signaling modulators, however, the evidence from figure 1 would suggest that this is not necessarily due to YrbE3A. Wouldn't the same trends be observed in the recombinant strains expressing only the empty vector?
A: Thanks a lot for your kind comments and question.
It would be best to set the control of the recombinant strains expressing only the empty vector when we checked the blocking effect by inhibitors of specific cell signaling modulators and we are sorry for this omission. However, we originally hoped use cytokine detection under the presence of inhibitors to confirm the results about the expression of phosphorylated signaling molecules, the latter having the negative controls of strains with vectors, and the cytokine results agreed with those of phosphorylated signaling molecules that JNK and p65 signal molecules but not Erk were involved in. Therefore, although this defect, we could keep our assertion based on the combination of findings from two parts. Overall, the NF-κB p65 and MAPK-JNK/p38 are involved in YrbE3A mediated production of TNF-α and IL-6 for Ms_YrbE3A; and NF-κB p65 and MAPK-p38 are involved in YrbE3A mediated production of TNF-α for BCG_YrbE3A.
To make the description more logical, would you please allow us to reverse the displaying order of these two parts, the original Fig.3A-B were changed to Fig.3C, while original Fig.3C to Fig.3A-B in the revised version. The text description was correspondingly modified. In addition, we discussed this defect in experimental design in the Discussion section. please see line 426-433.
Minor comments:
- The figures are so small that they are rather illegible. Also, several of the figures are difficult to decipher what is actually being presented.
A: Thanks a lot for this comment, and we are deeply sorry for the quality of our figures. We have now attached larger figures to the end of this letter. At the same time, we modified the size of the figures in our manuscript as much as possible.
- The figure legends must better describe what is being presented in the figures.
A: Thank you very much for your comments. As suggested, we checked all the legends and modified and believe the revised version is greatly improved.
- Figure 1a: A loading control is necessary for each of the samples. Also, how do we know whether there is actually a band somewhere on that gel that represents YrbE3A? A tagged fusion could be helpful in order to reveal protein expression on a Western Blot.
A: Thank you very much for your comments. To confirm the specific band of the recombinant YrbE3A, we firstly estimated the location of the band in SDS-PAGE gel in agreement with the expected molecular mass (32 kDa). Then we indeed performed Western blot assay. We separated the cellular proteins in cell lysis and proteins in bacterial membrane of Ms_YrbE3A and Ms_vec in SDS-PAGE and blotted to PVDF membrane. The blot was probed with his-tag monoclonal antibody (As shown in figure below, Lane 1 is the YrbE3A-cytoplasm, and lane 2 is YrbE3A-cytomembrane. Lane 3 and 4 are Vec-cytoplasm and Vec-cytomembrane, respectively). according to the molecular size of recombinant YrbE3A, we can conclude this YrbE3A expressed in the bacterial membrane. However, there are several non-specific bands in the blotted membrane probably the His-tag in this recombinant YrbE3A is too small. That is the reason why we didn’t show this picture in the original submitted version. We added this picture in revised paper as supplementary Fig.S1-B.
- Figure 1b: A quantitation of colony morphology would be helpful. Otherwise, it is difficult to see the differences in the colony morphology as printed. Also, how representative are these colonies of the recombinant strains? The pictures are small and, if printed in black and white, will be completely indiscernible.
A: Thank you for your valuable comments. Based on the pictures we took, we observed the 40 colonies for four strains and quantitates the diameters of colonies and numbers of wrinkles. The results are shown by the bar figure (Fig.1B). we added the description: “To test the effect of YrbE3A on the growth of recombinant strains, the colony morphology was observed. Generally speaking, the colonies of the YrbE3A expressing strains Ms_YrbE3A and BCG_YrbE3A were larger than their control strains Ms_Vec and BCG_vec (Fig.1A). By using ImageJ to measure the colonies, the circularity of Ms_YrbE3A and Ms_vec colonies was 0.36±0.05 and 0.71±0.01 respectively showing a signficantly difference (p<0.001). The average colony diameter of Ms_YrbE3A was significantly larger than that of Ms_vec (p<0.001) (Fig.1B). Similarly, the colony diameter and circularity of BCG_YrbE3A were larger than those of BCG_Vec (p<0.05) (Fig.1A and 1B).” please see line 206-212.
To make the pictures larger, would you please allow us to move the Fig.1A SDS-PAGE of recombinant YrbE3A, together with the extra western blot picture to supplementary Fig.S1 A and B.
- How do you know that YrbE3A is being expressed?
A: Thanks for your question. As described in the response to the above Comment 3, we checked the agreement of molecular mass of the expressed YrbE3A protein with our expected size by SDS-PAGE (Fig.S1 A) and reaction with the monoclonal antibody to His-tag by Western blot assay (Fig. S1B).
- Is yrbE3Aexpression must be controlled in figure 2 B-E.
A: Thanks for your question. It is very hard to control the expression of YrbE3A each time because the expression level is relatively low and the samples need to be concentrated before testing. However, we detected IL-6 expression in the supernatants of RAW264.7 cells treated with LPS (1μg/ml) alone and varying concentrations of recombinant YrbE3A protein purified from mc2155 (0, 2, 5, 10μg/ml) (As shown in below figure). According to the data, compared to LPS group, 5μg/ml and 10μg/ml YrbE3A can stimulate the highest expression of IL-6. We added this result into the Fig.2 as Fig.2B. Therefore, the sequence of current pictures in Fig.2B-E was changed into Fig.2C-2F.
Figure 2B. Detection of IL-6 expression specific to different concentrations of YrbE3A in infected macrophages.
- Line 214: IL-1beta does not appear to be significantly more expressed in Ms_yrbE3A versus Ms_Vec in figure 2b.
A: Thank you very much for your comments, and we agree with you that unlike TNF-α and IL-6, IL-1beta does not appear to be significantly more expressed in Ms_yrbE3A versus Ms_Vec although at mRNA level, the transcription of IL-1beta gene was significantly higher in Ms_yrbE3A than Ms_Vec (Fig.2D) . We added the description to clarify the differential expression of IL-1beta. please see line 257-261.
- In figure 2c, how different are the relative levels of mRNA expression between each of the two samples?
A: Thank you very much for your kind comment. We added the description of difference in the relative levels of mRNA expression between each of the two samples. please see line 262-266.
- How significant is the importance of YrbE3A for eliciting a pro-inflammatory response to the recombinant strain BCG_yrbE3A? The levels are not significantly different for any of the cytokines tested by ELISA except for TNF-alpha at 24h of infection.
A: Thanks a lot for your comments. Yes, as you said, in RAW264.7 cells infected with the recombinant strain BCG_yrbE3A, only TNF-α expression at protein level was significantly higher than that of the BCG_Vec group at 24h of infection (p<0.01). We added the statement in the text. please see line 257-266.
- Is purified YrbE3A sufficient to elicit a cytokine response or must it be exposed as a trans-membrane protein?
A: Thanks for your this smart comment. Yes, the purified YrbE3A is sufficient to elicit a cytokine response. We detected IL-6 expression in the supernatants of RAW264.7 cells treated with LPS (1μg/ml) as positive control and varying concentrations of purified recombinant YrbE3A protein (0, 2, 5, 10μg/ml) from the recombinant Ms_yrbE3A mentioned in above Comment 6. The effect ofYrbE3A protein at 5μg/ml on IL-6 production is significantly higher than LPS at 1μg/ml. We added this result in the text and Fig.2B.
- Does the expression of YrbE3A reflect the same level and kinetic profile as in TB?
A: Thanks for your kind question. It would be interesting to check the expression level and kinetic profiles in TB. However, we are sorry we didn’t check them because we didn’t make the antibody to YrbE3A during this study. We used antibody to His-tag to check the expression of YrbE3A in recombinant strains. We hope in the future we can clearly answer your question.
- If growth impacted by higher or lower levels of YrbE3A expression?
A: Thanks for your interesting question. It is a complex issue and we didn’t get a clear conclusion yet. However, since change in growth curves and colony shape and size was different between Ms_YrbE3A and BCG_Vec; and BCG_YrbE3A and BCG_Vec, we think that both growth characteristics of the parent strains and the expression levels of YrbE3A protein impact the growth of recombinant strains expressing YrbE3A proteins.
- Figure 3c does not seem to illustrate any difference in the expression of the cellular factors assayed. A quantitation of the levels of expression could be helpful in more easily interpreting these results.
A: Thanks for your kind comments. We agree with you because the pictures are too small, it is difficult to see the difference. Therefore, the quantitation of the levels of expression of Figure 3C (currently Fig.3A in revised version) was performed and the results were expressed in Figure 3D (currently Fig.3B in revised version).
- Figure 3d is so small that I can't see it well enough to judge whether the evidence is sufficient to support their claims.
A: Thank you very much for your kind comment. We are sorry for the small figures because we attempted to save the space and show all the data. To enlarge the pictures, we moved the legends to top and increased the picture size.
- Figure 4b shows that lymphocytes are initially more highly present than in the sample infected with Ms_Vec. Why does it flip later during infection?
A: Thanks for your careful reading and kind comment. As shown in Fig.4B, for the mice with MS_ YrbE3A infection, the lung lymphocytes expanded rapidly to the peak at day 2 post infection (PI), then gradually decreased at day 4 and 8 PI, dropped to the lowest at day 16 and 24 PI. It displayed a significant change over the time in agreement with natural history of infection. However, for the mice with MS_Vec infection, although the number of lymphocytes slightly increased after infection but this is an irregular fluctuation without significant difference over the time. Therefore, we think it means the number of lung lymphocytes in MS_ YrbE3A infection at day 16 and 24 dropped to the baseline which was similar to the level at day 0, not flipping over.
- The red arrows are poorly visible and the phenotype intended to be highlighted is not at all discernible. As such, I can't judge the merit of this figure.
A: Thank you very much for your kind comment. We are deeply sorry for picture quality. To avoid the confusion, we have deleted the arrows in Figure 4. Also, we increased the size of the figures.
- Figure 5 has panels that are also too small to be reliably judged.
A: Thank you very much for your kind comment. We are deeply sorry for the figure size. To make them better, we increased the size of line and bar charts by moving the side legends to the top or inside the charts, and some of tissue pictures to the supplementary figures.
Thank you very much for all of your comments and suggestions

Reviewer 2 Report
The manuscript entitled: "Mycobacterium tuberculosis YrbE3A promotes host innate immune response by targeting NF-κB/JNK signaling" by Jieru Wang, Xiaojie Zhu, Yongchong Peng, Tingting Zhu, Han Liu, Yifan Zhu, Xuekai Xiong, Xi Chen, Changmin Hu, Huanchun Chen, Yingyu Chen, and Aizhen Guo describes the role of Mycobacterium tuberculosis YrbE3A protein. The proposed model is focused on the modulation of inflammatory response triggered by YrbE3A expressed by Mycobacterium smegmatis and BCG in RAW 264.7 macrophages and mice. These studies indicate the YrbE3A as a potential therapeutic target.
Major issues:
- Fig.3: Did the Authors use the positive controls to show SC-514 and U-0126 to show the inhibition of the indicated signaling pathways in RAW 264.7 cells?
- This work requires extensive editorial and language correction.
Other comments:
3. Line 55: Is the term "toxicity" the right term used here?
4. Line 78: Please provide the name of NF-κB subunit (p65) for the antibodies used.
5. Line 79: cell signaling ---> Cell Signaling Technology?
6. Line 101:BCG_YrbE3A ---> BCG_Vec?
7. Line 139: Please provide secondary antibodies names
8. Line 153: 10% neutral PBS formalin ?
9. Line 188: "stagnation" phase ---> stationary phase
10. Line 189: Please check the font on Fig. 1C
11. Line 202: Ms_Vec/BCG_YrbE3A ---> Ms_Vec/BCG_Vec
12. Line 204: h post-infection (PI) ---> hpi
13. Line 243: Figure 3A: The legend requires proper positioning.
14. Line 243: Figure 3C: P65 ---> p65
15. Line 243: Figure 3C: p-JNK blot shows 11 lanes not 10.
16. Line 243: Figure 3D ---> NF-κ B ---> NF-κB
17. Line 263: INF-γ ---> IFN-γ
18. Line 272: Figure 4A requires correction. Pictures are not positioned correctly.
19. Line 273: Figure 4 description ("Hematoxylin and eosin (H&E) staining and lesions in lungs") is unclear and should be rewritten.
20. Line 291: Host innate immune ---> Host innate immune reponses?
21. Line 375: IL-α or IL-1α?
Overall, this work requires major revision.
Author Response
Dear reviewer,
Thanks very much for your comments concerning our manuscript entitled “Mycobacterium tuberculosis YrbE3A promotes host innate immune response by targeting NF-κB/JNK signaling” (ID: microorganisms-737544). In according with your advice and comments, we have made extensive modification on the original manuscript and added several extra data as required including:1.western blot assay of the recombinant YrbE3A; 2. the effect of purified YrbE3A at various concentration by using LPS on stimulation of the cytokine TNF-α; 3.the quantification of the colony size of the recombinant strains, etc. Revised portions are marked in red in the revised version of paper. The line number is followed by PDF format. In addition, the manuscript has been edited by the native English-speaking expert. We hope the revised version has been improved to meet with your requirement for publication consideration. The attached is our point-by-point response to your comments.
With best regards,
Aizhen Guo
The manuscript entitled: "Mycobacterium tuberculosis YrbE3A promotes host innate immune response by targeting NF-κB/JNK signaling" by Jieru Wang, Xiaojie Zhu, Yongchong Peng, Tingting Zhu, Han Liu, Yifan Zhu, Xuekai Xiong, Xi Chen, Changmin Hu, Huanchun Chen, Yingyu Chen, and Aizhen Guo describes the role of Mycobacterium tuberculosis YrbE3A protein. The proposed model is focused on the modulation of inflammatory response triggered by YrbE3A expressed by Mycobacterium smegmatis and BCG in RAW 264.7 macrophages and mice. These studies indicate the YrbE3A as a potential therapeutic target.
Major issues:
1.Fig.3: Did the Authors use the positive controls to show SC-514 and U-0126 to show the ion of the indicated signaling pathways in RAW 264.7 cells?
A: Thank a lot for your questions. We set the strains with an empty vector as the negative control in Western blot assay of signaling molecules, while TNF-α production as the positive control because it can be regulated by NF-κB and JNK signal pathways1-2.
2.This work requires extensive editorial and language correction.
A: Thanks a lot for this comment. We checked the whole manuscript carefully and performed extensive editorial and language correction. At the same time, we modified the size of the figures in our manuscript as much as possible, we have now attached larger figures to the end of this letter. Thus, the revised version should be improved a lot to meet your requirements.
Other comments:
- Line 55: Is the term "toxicity" the right term used here?
A: Thank you for your questions. We have replaced "toxicity" by "virulence".
- Line 78: Please provide the name of NF-κB subunit (p65) for the antibodies used.
A: Thanks a lot for this comment. We added NF-κB subunit (p65) for the antibodies.
- Line 79: cell signaling ---> Cell Signaling Technology?
A: Thanks a lot for your kind comment. We have changed Cell Signaling to Cell Signaling Technology in the revised version. please see line 75.
- Line 101: BCG_YrbE3A ---> BCG_Vec?
A: Thanks a lot for your kind comment and sorry for this incorrect expression. In the revised version, we corrected it, please see line 101-102.
- Line 139: Please provide secondary antibodies names
A: Thank for your comment. We added secondary antibodies names in the revised manuscript. please see line 146-151.
- Line 153: 10% neutral PBS formalin?
A: Thanks a lot for your kind comment. We changed to 10% neutral buffered formalin which was made with PBS (pH7.4, 0.1M) in the revised version. please see line 171-172.
- Line 188: "stagnation" phase ---> stationary phase
A: Thanks a lot for your kind comment. We corrected the word as suggested at line 217.
- Line 189: Please check the font on Fig. 1C
A: Thanks a lot for this comment. We corrected the font in Fig.1C in line 219.
- Line 202: Ms_Vec/BCG_YrbE3A ---> Ms_Vec/BCG_Vec
A: Thanks a lot for your kind comment. We correct it in line 243.
- Line 204: h post-infection (PI) ---> hpi
A: Thanks a lot for your kind comment and we are very sorry for our incorrect writing. We replaced “hpi” in the revised version and other places in the whole manuscript were added correspondingly.
- Line 243: Figure 3A: The legend requires proper positioning.
A: Thanks a lot for this comment. We correct the legend position.
- Line 243: Figure 3C: P65 ---> p65
A: Thanks a lot for your kind comment. We corrected it.
- Line 243: Figure 3C: p-JNK blot shows 11 lanes not 10.
A: Thanks for your comment. We are sorry to cause this confusion due to the improper image processing. Actually, to separate the different groups, we arranged an empty lane between them. To avoid the confusion, we cut off this lane in the revised version.
- Line 243: Figure 3D ---> NF-κ B ---> NF-κB
A: Thank you very much for comment. We corrected it, please see currently Fig.3A in revised version in line 301.
- Line 263: INF-γ ---> IFN-γ
A: Thank you very much for this comment. We are very sorry for our incorrect writing. We corrected it.
- Line 272: Figure 4A requires correction. Pictures are not positioned correctly.
A: Thank you very much for this comment. we corrected position.
- Line 273: Figure 4 description ("Hematoxylin and eosin (H&E) staining and lesions in lungs") is unclear and should be rewritten.
A: Thank you very much for your comment. We are very sorry for our incorrect writing. We corrected it as fellows: “YrbE3A expression promoted lung inflammatory lesion of mice infected with recombinant Ms_YrbE3A and Ms_Vec”. please see line 334.
- Line 291: Host innate immune ---> Host innate immune responses?
A: Thanks a lot for your kind comments and sorry for this misleading expression. As suggested, we used “host innate immune responses” instead. please see line 372.
- Line 375: IL-α or IL-1α?
A: Thanks a lot for your kind comment and we are very sorry for our incorrect writing. We corrected it in the whole manuscript were added correspondingly.
Thank you very much for all of your comments and suggestions
Reference:
- Hayden, M. S.; Ghosh, S., Regulation of NF-kappaB by TNF family cytokines. Semin Immunol 2014, 26 (3), 253-66.
- Sabio, G.; Davis, R. J., TNF and MAP kinase signalling pathways. Semin Immunol 2014, 26 (3), 237-45.

Round 2
Reviewer 1 Report
The authors have made significant effort to ameliorate the presentation of their manuscript and scientific evidence to support their claims. Some scientific points require additional clarification (points mentioned below). My most general concern is that the authors present a considerable ammount of information for which some cytokines go up and others don't depending on the samples, but also within individual samples at different time points. The results are complex, become confounding. Each individual result might be lost within the overall aims and conclusions to be drawn from this manuscript. I applaud the authors' efforts in putting together plenty of results. Some simplification in the figures or in the text might go a long way to clarifying main conclusions that are to be drawn from each of the figures.
The English should be further refined to correct grammatical mistakes, misspelled words, and to ameliorate the clarity of the text.
Scientific Points:
1) For macrophage infections in figure 2, it is not indicated in the figure legend or the materials and methods section (lines 100-110) whether IPTG was added to bacteria in order to stimulate the expression of YrbE3A in the recombinant strains. This point should be clarified especially since IPTG activation of gene expression might work differently in axenic conditions of growth versus during infection, where growing intracellular bacteria might exhibit limited accessibility to YrbE3A activation. Also, the nature of the "control" in figure 2b is not described in the legend or in the main text; this should be clarified.
2) I commend the authors' efforts to quantitate colony morphology. Nevertheless, the authors omit the definition of how circularity is quantitated and what the scale of the y-axis on figure 1B actually means. The figure legend should include the number of colonies analyzed for each sample. Also, the materials and methods section should either include a description of how ImageJ implements a measure of circularity or a reference describing how circularity is calculated.
3) Lines 347-349: The authors indicate that IL-6 levels are significantly increased in the YrbE3A recombinant strain. While the difference appears significant as per the authors' figure, the increase is likely less than a 20% increase. TNF-alpha exhibits a much greater difference in expression, as noted by the authors. These discrepencies represent a general allegory for how many results presented together may confound the main conclusions of each figure. One way to solve this problem is to add even more supplementary information: i.e. activate YrbE3A expression to different levels before/during infection and assay the level of cytokine secretion. Another way is to simplify the message down to a coherent core, with necessary supplementary information in support.
Author Response
Dear reviewer,
Thanks very much for your comments concerning our manuscript entitled “Mycobacterium tuberculosis YrbE3A promotes host innate immune response by targeting NF-κB/JNK signalling” (ID: microorganisms-737544). In according with your advice, we have made modification on the original manuscript by removing some pictures into the part of supplemental data and further editing the language. Revised portions are marked in red in the revised version of paper. The line number is followed by PDF format. In addition, the manuscript has been edited by the native English-speaking expert. We hope the revised version has been improved to meet with your requirement for publication consideration. The attached is our point-by-point response to the reviewer’s comments.
With best regards,
Aizhen Guo
The authors have made significant effort to ameliorate the presentation of their manuscript and scientific evidence to support their claims. Some scientific points require additional clarification (points mentioned below). My most general concern is that the authors present a considerable ammount of information for which some cytokines go up and others don't depending on the samples, but also within individual samples at different time points. The results are complex, become confounding. Each individual result might be lost within the overall aims and conclusions to be drawn from this manuscript. I applaud the authors' efforts in putting together plenty of results. Some simplification in the figures or in the text might go a long way to clarifying main conclusions that are to be drawn from each of the figures.
The English should be further refined to correct grammatical mistakes, misspelled words, and to ameliorate the clarity of the text.
Scientific Points:
(1)For macrophage infections in figure 2, it is not indicated in the figure legend or the materials and methods section (lines 100-110) whether IPTG was added to bacteria in order to stimulate the expression of YrbE3A in the recombinant strains. This point should be clarified especially since IPTG activation of gene expression might work differently in axenic conditions of growth versus during infection, where growing intracellular bacteria might exhibit limited accessibility to YrbE3A activation. Also, the nature of the "control" in figure 2b is not described in the legend or in the main text; this should be clarified.
A: Thank you very much for your kind comments. We are deeply sorry for the confusing writing.
Firstly, when we constructed recombinant strains, we used two vectors: pMV261 for MS and pET32a for E. coli (Rosetta and C43(DE3)). IPTG was added to pET32a of E. coli in order to stimulate the expression of YrbE3A, whereas pMV261- Ms_YrbE3A was the heat-inducible (Hsp60 promoter) with 45℃ water bath for 1h. However, when we wrote the original manuscript, we made a mistake in the description. We made corrections in the Materials and Methods section (YrbE3A expression was induced in a 45℃ water bath for 1h,) and Figure S1 legend (Lane 1, total Ms_YrbE3A cell lysate without heat-inducible; Lane 2, total Ms_YrbE3A cell lysate with heat-inducible). Moreover, when we performed recombinant strains to infect macrophages, these strains were not treated with a 45℃ water bath for 1h. It means, in intracellular bacteria, YrbE3A expression levels were not disturbed.
Secondly, we added the description of the "control" in Figure 2b: “RAW264.7 without LPS and YrbE3A protein served as control”
(2)I commend the authors' efforts to quantitate colony morphology. Nevertheless, the authors omit the definition of how circularity is quantitated and what the scale of the y-axis on figure 1B actually means. The figure legend should include the number of colonies analyzed for each sample. Also, the materials and methods section should either include a description of how ImageJ implements a measure of circularity or a reference describing how circularity is calculated.
A: Thanks a lot for your kind comments and question. We are deeply sorry for unclearly describe.
Firstly, according to the instructions of ImageJ, the circularity was calculated as 4π × area/perimeter2. A value of 1.0 indicates a perfect circle. As the value approaches 0.0, it indicates an increasingly elongated shape. (https://imagej.nih.gov/ij/docs/menus/analyze.html#set). We also added it in Materials and Methods section. Please see line 98-100.
Secondly, we have added the number of colonies analyzed for each sample into the figure legend: “Ten independent colonies were analyzed for each strain.” Please see line 230.
Thirdly, we have added “In order to describe the morphology of the four strains, the colony circularity and diameter of four recombinant strains were measured by ImageJ software according to the instructions 1. Ten independent colonies were analyzed for each strain.”. Please see line 95-97 and 238.
The reference is: “Zhi, L. , Ling-Ling, W. , Wei-Dong, Z. , Yi-Fang, C. , & Zhong, W. . (2011). The surface of the geometric characteristics analysis for rice endosperm starch granules by using image j. Journal of Chinese Electron Microscopy Society, 30, 466-471.”.
(3)Lines 347-349: The authors indicate that IL-6 levels are significantly increased in the YrbE3A recombinant strain. While the difference appears significant as per the authors' figure, the increase is likely less than a 20% increase. TNF-alpha exhibits a much greater difference in expression, as noted by the authors. These discrepencies represent a general allegory for how many results presented together may confound the main conclusions of each figure. One way to solve this problem is to add even more supplementary information: i.e. activate YrbE3A expression to different levels before/during infection and assay the level of cytokine secretion. Another way is to simplify the message down to a coherent core, with necessary supplementary information in support
A: Thank you for your valuable comments.
Firstly, as suggested, we checked all the statistics analysis for the results again. At day 8 postinfection with Ms_Vec and Ms_YrbE3A, the expressions of IL-6 are 48.86 ±2.98 and 80.81±9.88 pg/mL, and the expressions of TNF-α are 31.07±1.9 and 44.92±2.61 pg/mL, respectively. Using two-way ANOVA followed by LSD test, we found there were significant differences in lung IL-6 and TNF-α expressions between Ms_Vec-infected mice and Ms_YrbE3A-infected mice at day 8.
Secondly, we simplified the figures down to a coherent core by moving some data into part of supplementary information. Please see Figure 2, 4, 5, S2, S3 and S4.
Thank you very much for all of your comments and suggestions

Reviewer 2 Report
The manuscript entitled: "Mycobacterium tuberculosis YrbE3A promotes host innate immune response by targeting NF-κB/JNK signaling" by Jieru Wang, Xiaojie Zhu, Yongchong Peng, Tingting Zhu, Han Liu, Yifan Zhu, Xuekai Xiong, Xi Chen, Changmin Hu, Huanchun Chen, Yingyu Chen, and Aizhen Guo describes the role of Mycobacterium tuberculosis YrbE3A protein. The proposed model is focused on the modulation of inflammatory response triggered by YrbE3A expressed by Mycobacterium smegmatis and BCG in RAW 264.7 macrophages and mice. These studies indicate the YrbE3A as a potential therapeutic target.
There are some minor modifications needed:
Line 38: signal pathways --> signaling pathways; cytokines --> proinflammatory cytokines
Line 464: M.tb ---> Mtb; M.bovis ---> the abbreviations has not been introduced to the text of the manuscript
Author Response
Dear reviewer,
Thanks very much for your comments concerning our manuscript entitled “Mycobacterium tuberculosis YrbE3A promotes host innate immune response by targeting NF-κB/JNK signalling” (ID: microorganisms-737544). In according with your advice, we have made modification on the original manuscript by removing some pictures into the part of supplemental data and further editing the language. Revised portions are marked in red in the revised version of paper. The line number is followed by PDF format. In addition, the manuscript has been edited by the native English-speaking expert. We hope the revised version has been improved to meet with your requirement for publication consideration. The attached is our point-by-point response to the reviewer’s comments.
With best regards,
Aizhen Guo
The manuscript entitled: "Mycobacterium tuberculosis YrbE3A promotes host innate immune response by targeting NF-κB/JNK signaling" by Jieru Wang, Xiaojie Zhu, Yongchong Peng, Tingting Zhu, Han Liu, Yifan Zhu, Xuekai Xiong, Xi Chen, Changmin Hu, Huanchun Chen, Yingyu Chen, and Aizhen Guo describes the role of Mycobacterium tuberculosis YrbE3A protein. The proposed model is focused on the modulation of inflammatory response triggered by YrbE3A expressed by Mycobacterium smegmatis and BCG in RAW 264.7 macrophages and mice. These studies indicate the YrbE3A as a potential therapeutic target.
There are some minor modifications needed:
Line 38: signal pathways --> signaling pathways; cytokines --> proinflammatory cytokines
A: Thanks a lot for your kind comments. We corrected them as suggested, please see line 40.
Line 464: M.tb ---> Mtb; M.bovis ---> the abbreviations has not been introduced to the text of the manuscript
A: Thank you very much for this comment. We change M.tb to Mtb and added the abbreviation of M.bovis in the text of the manuscript. please see line 50, 52-54.
Thank you very much for all of your comments and suggestions
